# Immunomodulatory and Antioxidant Drugs in Glaucoma Treatment

**DOI:** 10.3390/ph16091193

**Published:** 2023-08-22

**Authors:** Francesco Buonfiglio, Norbert Pfeiffer, Adrian Gericke

**Affiliations:** Department of Ophthalmology, University Medical Center, Johannes Gutenberg University Mainz, Langenbeckstrasse 1, 55131 Mainz, Germany; norbert.pfeiffer@unimedizin-mainz.de

**Keywords:** autoimmune, glaucoma, retinal ganglion cell, optic nerve, inflammation, ischemia, oxidative stress, vascular dysregulation

## Abstract

Glaucoma, a group of diseases characterized by progressive retinal ganglion cell loss, cupping of the optic disc, and a typical pattern of visual field defects, is a leading cause of severe visual impairment and blindness worldwide. Elevated intraocular pressure (IOP) is the leading risk factor for glaucoma development. However, glaucoma can also develop at normal pressure levels. An increased susceptibility of retinal ganglion cells to IOP, systemic vascular dysregulation, endothelial dysfunction, and autoimmune imbalances have been suggested as playing a role in the pathophysiology of normal-tension glaucoma. Since inflammation and oxidative stress play a role in all forms of glaucoma, the goal of this review article is to present an overview of the inflammatory and pro-oxidant mechanisms in the pathophysiology of glaucoma and to discuss immunomodulatory and antioxidant treatment approaches.

## 1. Introduction

The word glaucoma subsumes a spectrum of disorders, which share a progressive optic nerve atrophy derived from the loss of retinal ganglion cells (RGCs), with concomitant optic disc cupping, retinal nerve fiber layer (RNFL) thinning, and clinically detectable early visual field losses in the form of arcuate defects that correspond to the fiber nerve bundle pattern [1,2,3,4]. Subsequently, in late disease stages, advanced optic nerve atrophy and perimetric defects can ultimately lead to blindness. Glaucoma is among the leading causes of irreversible visual loss worldwide [5,6,7,8]. Elevated intraocular pressure (IOP) is the major risk factor for this disorder [9,10]. Nonetheless, IOP alone appears not to be sufficient to properly account for all cases of glaucoma, since this disease can also occur without an elevation of IOP, such as in cases of normal-tension glaucoma (NTG) [11]. Relatively recent research has shed light on the multifaceted pathophysiology of glaucoma, collecting evidence about the involvement of vascular dysfunction, an altered redox status, neuroinflammation, and autoimmunity as additional actors in glaucomatous pathogenesis [11,12,13,14,15,16,17]. Considering the overall high prevalence and severity of this disorder, various publications have underlined the need for effective therapeutic strategies, exploring new pharmaceutical fields for glaucoma, with the purpose of preventing the severe visual impairment that occurs in the late stages [18,19,20].

A profound comprehension of the pathophysiological events in glaucoma is propaedeutic for eventually considering new alternative targets, which may finally hold additional benefits for patients. In this regard, this work aims to summarize the current understanding of the complex glaucomatous etiopathogenesis, highlighting alternative insights related to emerging pathomechanisms, such as inflammation and oxidative stress. Furthermore, we will explore immunomodulatory and antioxidant proposals as effective curative options for glaucoma.

## 2. General Characteristics of Glaucoma

### 2.1. Classification, Epidemiology, and Economic Implications

Glaucoma is classified into primary and secondary forms based on the presence or absence of pre-existent pathological conditions such as uveitis, neoangiogenesis, traumas, and lens abnormalities [2,21]. Additionally, glaucoma can be categorized as either open-angle or angle-closure, based on the chamber angle located between the iris and the posterior surface of the cornea [22]. In a healthy state, this angle is physiologically open, allowing the outflow of aqueous humor (AH) through the trabecular meshwork (TM) to the uvea and conjunctiva, maintaining normal turnover [23,24]. Primary open-angle glaucoma (POAG) is the most common form of glaucoma [7], and is often associated with high IOP. However, it also includes a subtype known as normal-tension glaucoma (NTG), in which the IOP is not elevated. NTG accounts for 30–90% of POAG cases and its prevalence varies significantly depending on geographical location [11,25]. Possible explanations for this significant difference have been attributed to an alternative risk-factor profile found in different populations, such as genetic components [26], long axial length [27], low intracranial pressure, and vascular dysregulation [25].

In the context of primary angle-closure glaucoma (PACG), there is anatomical contact between the iris and the cornea, and in 90% of cases, between the iris and the lens, creating a pupillary block [28]. Although PACG cases account for approximately 26% of total glaucoma cases [29], they are responsible for approximatively half of worldwide cases of glaucoma-related blindness [6,30].

From an epidemiological standpoint, it was estimated that in 2013, approximately 64.3 million people between the ages of 40 and 80 were affected by glaucoma. However, by the year 2040, it is projected that the number of individuals affected will exceed 110 million [7]. The direct costs associated with this condition are primarily linked to disease progression and the need for treatment adjustments when initial therapies are unsuccessful, contributing to cost escalation [31]. Indirect costs, such as the loss of well-being and visual disability experienced by patients, have been estimated to be the most impactful economic factors in Europe [32]. Additionally, glaucoma has been shown as a possible risk factor for falls which require hospitalization [33]. Collectively, numerous studies on this subject have emphasized the importance of halting disease progression and preventing late-stage glaucoma to minimize the loss of well-being for patients and to prevent escalating costs.

Glaucomas are chronic and progressive optic neuropathies that, if left untreated, can potentially lead to irreversible visual loss. According to the existing literature, 15 to 20% of patients with glaucoma may experience unilateral blindness [34,35,36,37]. The prognosis can vary depending on the subtype of glaucoma [6,38]. However, it is important to note that the majority of patients with glaucoma can maintain useful vision through appropriate treatments aimed at lowering IOP [39,40]. The early detection and management of glaucoma, along with regular follow-ups and adherence to treatment regimens, play a crucial role in preserving vision and delaying disease progression.

### 2.2. Symptoms and Diagnostic Features

POAG and PACG present with different sets of symptoms. In POAG, the disease progression is often asymptomatic due to binocular compensation. As a result, patients typically experience the first noticeable symptoms only in advanced stages when significant damage to the visual field has already occurred [22,41].

On the other hand, PACG manifests with rapid and painful symptoms. Affected individuals may experience a rock-hard sensation in the eye, corneal edema, reduced visual acuity, conjunctival hyperemia (redness), irradiating pain, and potentially accompanying nausea and vomiting [22]. PACG is considered an ophthalmologic emergency that necessitates immediate medical intervention to prevent severe visual loss [22].

The use of appropriate diagnostic tools is crucial for facilitating the detection of early signs of glaucoma and initiating prompt and appropriate therapy to prevent further damage. Tonometry, fundoscopy, and perimetry are valuable in enabling an early diagnosis [22,42,43]. Classic fundoscopic signs of glaucoma include an enlarged optic cup, resulting in an increased cup–disc ratio, the loss of the neuroretinal rim, the presence of disc hemorrhages, and parapapillary tissue atrophy [22,43].

Assessing the disease progression of glaucoma can be achieved through an examination of the neuroretinal rim of the optic nerve head using fundoscopy or through perimetric evaluation [44]. Recently, SD-OCT has also been described as a suitable diagnostic tool for staging glaucoma [45]. However, despite the availability of various diagnostic features for assessing glaucomatous disease progression, there is currently no consensus on a singular criterion to determine the specific disease stages [46].

### 2.3. Pharmaceutical Approaches to Treatment and Surgical Interventions

The primary objective of the major established antiglaucoma drugs is to reduce IOP to a personalized and acceptable range to halt the progression of the disorder [47]. These medications are typically administered topically via eye drops, and can be categorized based on their pharmacological mechanisms into the following groups:Prostaglandin analogues: examples include bimatoprost, which enhances both trabecular and uveoscleral outflow of AH [22].β-blockers: medications like levobunolol and timolol work by reducing the production of AH [22].α_2_-adrenoceptor agonists: drugs such as apraclonidine and brimonidine lower IOP by decreasing aqueous humor production and augmenting trabecular outflow [22].Carbonic anhydrase inhibitors: agents like brinzolamide act by reducing the production of aqueous humor [22].Miotic agents: pilocarpine, for instance, increases the chamber angle by constricting the pupil and can also provide neuroprotective effects through the activation of muscarinic receptors [48].Rho-associated protein kinase (ROCK) inhibitors: Netarsudil is a ROCK inhibitor that targets the ROCK pathway, suppressing fibrotic events in the trabecular meshwork ™ and optimizing aqueous humor flow, thereby reducing IOP [49]. This molecule has been approved for the treatment of glaucoma in the United States (2017) and Europe (2019) in the form of a 0.02% ophthalmic topical formulation for once-daily application [50].

Additionally, laser and surgical procedures are well-established in clinical practice for the management of both open-angle and angle-closure glaucoma. These therapeutic options have the goal of improving the outflow of AH or to reduce its production [4,22,51,52]. Surgical interventions in glaucoma are usually considered a second-line therapy when conservative options fail to sufficiently lower IOP. These surgical options are, for example, cyclocryocoagulation, minimally invasive procedures such as stent implantation, and filtering procedures like trabeculectomy [22]. Due to the scarring processes that may affect the long-term efficacy of surgical techniques bypassing the outflow of AH to subconjunctival spaces, medications are often employed postoperatively to inhibit excessive scar tissue growth [53]. Commonly used medications for this purpose in clinical practice include topical steroids and non-steroidal anti-inflammatory drugs. Off-label drugs, such as 5-fluorouracil and mitomycin C, are also utilized [53]. Additionally, there are ongoing investigations into the use of biologic drugs, such as bevacizumab (anti-vascular endothelial growth factor, VEGF), as well as molecules targeting the transforming growth factor (TGF-β) signaling pathway, like lerdelimumab (anti-TGF-β2) and decorin (a proteoglycan that also targets TGF-β signaling) [53].

## 3. Pathophysiology

Based on the existing literature, we will now discuss the primary etiological factors and subsequent biomolecular pathomechanisms that lead to RGC loss and optic nerve head (ONH) atrophy in glaucoma.

### 3.1. Risk Factors

#### 3.1.1. Elevated Intraocular Pressure

The main risk factor for both POAG and PACG is an elevated IOP, defined as a pressure value above the 97.5th percentile for the specific population under consideration, often considered to be higher than 21 mmHg [2,54]. In addition to IOP, other risk factors for POAG include myopia, advanced age, belonging to the black ethnic group, and a family history of the condition. For PACG, risk factors include being female, having a small corneal diameter, hyperopia, an anteriorly positioned lens, and shallower central and limbal anterior chamber depth [29]. However, IOP is recognized as the primary modifiable risk factor, making it the main target of current established antiglaucoma drugs [22].

Two major theories have been proposed to explain the pathogenesis of glaucoma, both emphasizing the association between an elevated IOP and the development of the disease: the “vascular” and the “mechanical” theories. According to the vascular theory, a high IOP leads to compression of the blood vessels supplying the ONH, resulting in reduced blood flow, hypoperfusion, and subsequent ischemia in RGCs [55,56]. On the other hand, the mechanical theory suggests that an elevated IOP causes compressions and deformations of the lamina cribrosa and RGC axons, initiating a cascade of events that lead to cell death due to blocked axoplasmic traffic and inadequate cellular supply [57]. Figure 1 provides a summary of the events leading to mechanical damage in RGCs as the consequence of an elevated IOP.

An elevated IOP is proposed to arise from a pathological increase in resistance to AH flow within the TM [58]. The TM, located in the chamber angle, consists of three layers: the uveal TM, corneoscleral TM, and the juxtacanalicular TM (also known as the cribriform TM region), which borders Schlemm’s canal. AH flows through the TM and reaches the episcleral veins of the conjunctiva via the Schlemm’s canal [59]. The permeability of the TM to AH plays a crucial role in regulating IOP levels [59]. Structural alterations in the TM can lead to the apoptosis of TM cells and the disintegration of its structure [60]. Additionally, changes in the deposition of the extracellular matrix within the TM can disrupt the adhesion of TM-endothelial cells [61]. TGF-β2 appears to play a pivotal role in promoting the deposition of the extracellular matrix within the human TM during glaucoma [62]. These events ultimately result in increased resistance to AH drainage within the TM, leading to an elevated IOP [60].

#### 3.1.2. Genetic Factors, Systemic Vascular Dysregulation, and Endothelial Dysfunction

In the “Collaborative Normal-Tension Glaucoma Study”, a clinical trial, the effectiveness of IOP-lowering therapy in NTG was evaluated. The study revealed that although reducing IOP can have a positive impact, it alone cannot completely halt disease progression [12,63]. This suggests the involvement of additional factors in the development of NTG. The wide geographical variability in the prevalence of NTG and the relatively high percentage (approximately 21%) of patients reporting a family history of the condition [64] suggest a possible genetic predisposition. Individuals with NTG may have a lower tolerance for what are considered “normal” IOP levels [12]. The increased susceptibility of RGCs to IOP-induced damage is believed to contribute to the mechanical injuries observed in NTG, similar to those seen in glaucomas associated with an elevated IOP [12]. Numerous specific gene polymorphisms, resulting in the altered functionality of corresponding proteins, have been associated with NTG [64]. For example, certain sequence variants of the optineurin (OPTN) gene, which encodes a neuroprotective and IOP-regulating protein, have been linked to NTG [65]. Zhu and colleagues have provided evidence supporting the neuroprotective role of OPTN in RGCs by counteracting inflammation and apoptosis. They found that OPTN negatively regulates the tumor necrosis factor-α (TNF-α)-induced NF-κB activation, which plays a crucial role in cell survival [66]. Additionally, Minegishi et al. extensively reviewed the significance of OPTN in glaucoma [67]. Specifically, they focused on the most common OPTN mutation in NTG, known as E50K, and highlighted its impact in triggering abnormal aggregation of intracellular vesicles [68,69]. Moreover, the E50K mutation was associated with a disruption of the Golgi structure, leading to cellular toxicity [70,71,72,73]. In addition, mutations in the optic atrophy type 1 (OPA1) gene, which is crucial for mitochondrial dynamics, have also been implicated in the pathogenesis of NTG. These mutations can lead to RGC apoptosis through mitochondrial dysfunction [64,74]. Furthermore, specific gene sequence variants of the endothelin-1 (ET-1) receptor A have been identified as being associated with NTG [75].

In addition to genetic factors, several alternative risk factors have been identified as potential contributors to the pathophysiology of NTG, including systemic vascular dysregulation, oxidative stress, and endothelial dysfunction [11,12,17]. A systemic vascular impairment, such as cerebral silent infarcts and nocturnal arterial hypotension, has been associated with NTG, potentially leading to the condition of hypoperfusion in the ONH [11,12,76,77,78]. Hypoperfusion-induced hypoxia may initiate the glaucomatous pathogenesis of NTG [79]. As a result of hypoxic insults, the hypoxia-inducible factor 1α (HIF-1α), a potent cytokine, triggers downstream transductions that activate glial cells, leading to neuroinflammation, similar to the events observed in glaucoma associated with a high IOP [14,80,81]. Vascular endothelial dysfunction is another characteristic of NTG and may manifest through the impairment of vasoregulatory factors, such as nitric oxide (NO) [82] and ET-1 [64]. Excessive reactive oxygen species (ROS) can reduce NO-dependent vasorelaxation due to the impaired activity of endothelial nitric oxide synthase (eNOS). In the context of an altered redox status, the fundamental cofactor of eNOS, tetrahydrobiopterin, undergoes oxidation to dihydrobiopterin, resulting in abnormal eNOS activity, the production of peroxynitrite (ONOO^−^), and a lower bioavailability of NO [82]. Consequently, dysfunctional vasoregulation occurs, leading to deficits in vasorelaxation [64,83,84]. Moreover, the vasoconstrictor peptide ET-1 has been reported to be increased in the plasma [85,86] and in the AH [87] of NTG patients. The abnormal vasoconstriction induced by ET-1 may affect the blood vessels supplying the ONH in NTG, further contributing to reduced perfusion [88]. The combined processes of decreased NO-dependent vasodilation and increased ET-1-induced vasoconstriction in blood vessels may result in a reduced perfusion of the ONH, forming the etiopathogenic basis for primary damage to RGCs in NTG [64,89].

Figure 2 illustrates the pathomechanisms due to increased susceptibility to IOP in RGCs, to hypoperfusion, and to endothelial dysfunction in NTG.

### 3.2. Pathomechanisms

#### 3.2.1. Chronic Oxidative Stress

Our own studies in mice have demonstrated that an elevated IOP leads to endothelial damage in retinal blood vessels and the upregulation of nicotinamide adenine dinucleotide phosphate oxidase type 2 (NOX-2), a major enzyme responsible for generating ROS from oxygen (O_2_) in the form of superoxide (O_2_^•−^) [90,91]. Similar findings were observed in a rat model of glaucoma, where chronic high IOP-induced hypoperfusion of the ONH, axonal transport impairment, and excessive ROS production through the upregulation of NOX-2 occurred [92]. Hypoxia, which occurs during low oxygen concentrations, can also contribute to ROS production as the electron transport chain slows down, leading to an accumulation of reducing equivalents and subsequent O_2_^•−^ production [92]. Hypoxia-inducible factor 1-alpha (HIF-1α) is upregulated in response to hypoxia in glaucoma patients and can further enhance NOX-2 and inducible nitric oxide synthase (iNOS) expression, resulting in ROS production [13,93,94,95,96]. ROS, in turn, can trigger the expression of HIF-1α, creating a feedback loop that amplifies inflammation and apoptosis [97,98]. Glial activation and the release of TNF-α follow, leading to the activation of the nuclear factor kappa-light-chain-enhancer of activated B cells (NF-kB), a transcription factor responsible for inflammation. This process amplifies glial activation, neuroinflammation, and ultimately apoptosis [14,80,81]. The activation of the apoptosis signal-regulating kinase 1 (ASK-1)/p38 mitogen-activated protein kinase (MAPK)/JNK/extracellular-signal-regulated kinase (ERK) axis by ROS can lead to caspase-3 activation and cellular membrane disassembly, promoting cell death [99,100]. Chronic exposure to ROS can activate the phosphoinositide 3-kinase (PI3K)/Akt axis while attenuating the mammalian target of rapamycin (mTOR) pathway, further stimulating the NF-kB and enhancing inflammatory events [101]. Additionally, excessive ROS disrupts glutamate metabolism, leading to the neurotoxic extracellular accumulation of glutamate, as dysfunctional glial cells fail to properly buffer the excess glutamate [16,102,103]. Moreover, oxidized metabolites, like advanced glycation end products (AGEs) and oxidized low-density lipoproteins (oxLDLs), can act as “antigenic” stimuli, promoting ROS production, NF-kB activity, glial activation, and apoptosis [15,16].

#### 3.2.2. Mitochondrial Dysfunction

Mitochondrial dysfunction plays a central pathophysiological role in glaucoma and is associated with inflammation, oxidative stress, impaired mitochondrial dynamics and reduced ATP production [104]. Excessive ROS and metabolic stress due to a nutrient deficit can lead to mtDNA mutations and subsequent mitochondrial dysfunction [105]. Mechanical insults from an elevated IOP can cause mitochondrial disruptions and deficiencies in the OPA1 gene, which regulates mitochondrial fusion, a process important for mitochondrial quality control [15,104,106,107,108]. A deficiency in OPA1 can trigger ROS overabundance and glutamate excitotoxicity [109]. Conversely, the upregulation of OPA1 has been shown to have a protective effect on RGCs by enhancing mitochondrial fusion and mitophagy, the selective autophagy of damaged mitochondria [110,111,112]. In glaucoma, the balance between mitochondrial fusion and fission is disrupted, leading to increased fission and reduced fusion and mitophagy, which results in an elevated mitochondrial number and decreased mitochondrial size [113,114,115]. In humans, mitochondrial fission is mainly mediated by the dynamin-related protein 1 (Drp-1) [116]. In a murine model of glaucoma, an elevated IOP leads to the dephosphorylation of Drp-1, resulting in mitochondrial fragmentation and RGC loss via apoptosis [117]. A recent in vitro study demonstrated that an ERK1/2-Drp1-ROS axis induced by an elevated IOP could trigger mitochondrial dysfunction and apoptosis in RGCs [118]. Furthermore, oxidized mitochondrial DNA and mitochondrial fragments released from microglia can activate the NLRP3 inflammasome, leading to the production of inflammatory cytokines [15,106,119]. Mitochondria are also involved in glial neuroinflammation processes through the mitochondrial ROS-generated activation of NF-κB, leading to the production of inflammatory cytokines [15].

#### 3.2.3. Endoplasmic Reticulum Stress

The endoplasmic reticulum (ER) and mitochondria interact through calcium-dependent processes, influencing each other and leading to energy deficiency, apoptosis, inflammation, and increased ROS production [102]. The ER is an intracellular organelle responsible for protein processing and folding to ensure their proper functionality [120,121]. Various conditions such as oxidative stress, protein mutations, viral infections, nutritional deficits, and hypoxia can impact the ER, leading to an accumulation of unfolded proteins [122,123,124]. This results in ER stress, triggering the unfolded protein response (UPR) to restore cellular homeostasis [125]. Chronic ER stress can paradoxically perpetuate UPR activation, leading to apoptosis, the activation of NF-kB, and further ROS formation [123,124]. The UPR consists of three main signaling pathways:The inositol-requiring protein 1 (IRE-1)/spliced X-box binding protein-1 (sXBP1)/Janus Kinase (JNK) pathway, which improves protein folding but can also induce inflammation and apoptosis [120,124,126].The protein kinase RNA-like endoplasmic reticulum kinase (PERK)/eukaryotic initiation factor 2α (eIF2α)/activating transcription factor 4 (ATF4)/CCAAT-enhancer-binding protein homologous protein (CHOP) pathway, which reduces protein translation but can increase ROS production and promote apoptosis [127].The activating transcription factor 6 (ATF-6) pathway, which enhances the elimination of misfolded proteins and optimizes protein folding but may also activate proapoptotic cascades [120,124,128].

The ROS generated during ER stress, particularly through the ATF4/CHOP pathway, can activate inflammasomes, leading to increased neuroinflammation and further damaging mitochondria [127].

#### 3.2.4. Neuroinflammation and Glial Activation

Elevated hydrostatic pressure and ischemia can trigger the release of the major proinflammatory cytokine, TNF-α, from the glial cells, initiating inflammation and apoptosis in RGCs [80]. TNF-α plays a pivotal role in glaucomatous inflammation and oxidative processes. It is secreted by microglia, astrocytes, and Müller cells and contributes to mitochondrial dysfunction, increased ROS levels, and NF-kB expression, which in turn promote the expression of proinflammatory cytokines and adhesion molecules [14,80]. Heat shock proteins (HSPs) and mitochondrial damage-associated molecular patterns (DAMPs) have been investigated as “highly antigenic molecules” associated with neuroinflammation in glaucoma [129,130,131,132]. These molecules can activate the Toll-like receptors (TLRs) expressed in glial cells, leading to NF-kB activation and neuroinflammation [72,130]. The dysregulation of the complement system and the infiltration of activated T cells and monocytes have also been implicated in RGC death [129,133,134,135,136]. Indeed, studies conducted on murine glaucoma models have provided evidence that the absence of the complement can attenuate disease progression [137,138]. In recent years, a novel process called necroptosis has been introduced as the mechanism responsible for axonal degeneration in neurodegenerative disorders. This process can be triggered by TLR-, Fas-, TNF-α-, and interferons (IFNs)-signaling, and is characterized by cell swelling, granular cytoplasm, and cellular lysis [139]. Unlike apoptosis, which typically involves caspases, necroptosis relies on kinase-mediated transductions [140]. Importantly, apoptotic cell death is generally immunosuppressive, while necroptotic cell death triggers inflammation [140]. Ko and co-workers recently demonstrated in an experimental neuroinflammatory model of glaucoma that TNF-α can exclusively induce necroptosis in axons. This process is dependent on the presence of sterile alpha and TIR motif 1 (SARM1), and involves the reduction of axon survival factors, such as nicotinamide mononucleotide adenylyltransferase 2 and stathmin 2. Additionally, the activation of SARM1 NADase leads to calcium influx and subsequent axon degeneration [141].

Collectively, new insights into the neuroinflammatory processes highlight the role of microbiota via TLR-signaling and of specific programmed cell death pathways, like SARM1-dependent necroptosis, which require a more complete understanding to possibly transfer this new knowledge into the design of experimental immunomodulatory strategies.

#### 3.2.5. Autoimmune Imbalance

Evidence of autoimmune factors in glaucoma has been described, with autoantibodies detected in the sera and retina of glaucoma patients [17,142,143,144,145,146,147,148,149,150,151]. Heat shock proteins (HSP) may play a critical role in this context. HSPs can be produced by bacteria or generated endogenously by cells at the sites of inflammation, and they can activate specific HSP-induced T-regulatory cells [152]. High levels of HSP autoantibodies, including antibodies against HSP27, have been found in the sera of glaucoma patients. These autoantibodies have been shown to trigger neuronal apoptosis by interfering with the function of native HSPs in stabilizing the cytoskeleton [153,154,155]. Autoantibodies against HSP60 [156] and HSP70 [157] have also been detected in the sera of glaucoma patients [158]. Furthermore, studies have demonstrated IgG autoantibody depositions in glaucomatous retinas, along with an increase in CD27+/IgG+ plasma cells and elevated levels of TNF-α, IL-6, and IL-8. These proinflammatory mediators were found to be released by activated microglia [142].

On the other hand, glaucoma patients have shown the downregulation of protective, naturally occurring autoantibodies against 14-3-3 and γ-synuclein, which may contribute to secondary injuries in RGCs [158,159].

Taken together, and considering the sequence of the pathogenetic events, imbalances between pro-apoptotic and anti-apoptotic autoantibodies in autoimmune responses may contribute to secondary injuries in RGCs [158,160]. The autoantibodies found in glaucoma patients may serve as useful diagnostic biomarkers [161].

Figure 3 summarizes the processes leading to the loss of RGCs in glaucoma.

## 4. Emerging Curative Strategies: Immunomodulatory and Antioxidants

In the current glaucoma research, two main branches can be distinguished. The first area focuses on the implementation of established and commonly used antiglaucoma drugs, which target factors like aqueous humor production and outflow pathways to lower the IOP. The second area is a more recent and emerging field with the goal of testing and developing neuroprotective approaches to prevent or mitigate RGC loss. In the following sections, we explore the potential therapeutic options within the realms of immunomodulation and antioxidants, which have been proposed as neuroprotective strategies.

### 4.1. Immunomodulatory Candidates for Glaucoma

A molecule that has garnered significant interest in antiglaucomatous explorations, due to its immunomodulating features, is the fragment apoptosis stimulator (Fas) ligand. This membrane-bound protein has been found in the eye and is known to exhibit pro-inflammatory and pro-apoptotic activity when it binds to its receptor [162]. However, when it is cleaved and released as a soluble isoform, Fas exhibits the opposite functions [163]. A study conducted on a mouse model of glaucoma demonstrated that an upregulation of the soluble form of Fas ligand, achieved through intravitreal adeno-associated virus-mediated gene treatment, can reduce glial cell activation and prevent the loss of RGCs [164]. Moreover, a small peptide inhibitor of the Fas receptor, known as ONL1204, has shown promising results in a murine glaucoma model by suppressing RGC apoptosis, preserving axons, and inhibiting glial activation and neuroinflammation [165]. A dedicated clinical trial is currently underway to evaluate the effectiveness of an ophthalmic solution of ONL1204 on 25 patients with progressing open-angle glaucoma (NCT05160805). The estimated completion date of this study is September 2023.

Another class of molecules with immunomodulating characteristics is represented by the adenosine receptor modulators. For example, caffeine, by antagonizing the adenosine A2A receptor, has been shown to have the ability to protect against neuroinflammation and attenuate glial activation in neurodegenerative conditions [166]. Building on this evidence, studies have investigated the effectiveness of selective A2A receptor antagonists, such as SCH 58261 in an animal model of ischemia/reperfusion [167], as well as caffeine in a rodent model of glaucoma [168]. In both cases, these interventions demonstrated a reduction in neuroinflammation through decreased glial activation, ultimately resulting in the preservation of RGCs. In addition, caffeic acid phenethyl ester, when administered in a rodent model of glaucoma, has been shown to reduce the expression of pro-inflammatory cytokines, such as IL-6 and IL-8, as well as inducible nitric oxide synthase (iNOS) and COX2. This leads to a decreased activation of NF-kB, thereby attenuating neuroinflammation and preventing RGC loss [169].

A modulator of the adenosine A3 receptor called FM101 has been demonstrated to be safe in a rodent model of glaucoma [170], and a dedicated clinical trial is currently underway to evaluate its efficacy in patients with ocular hypertension (NCT04585100).

Biologics have also been tested as potential pharmacological options in glaucoma for their ability to modulate inflammation. For example, etanercept is a monoclonal antibody that targets and antagonizes the human TNF-α receptor type 2 [18]. This biologic drug is approved for the treatment of autoimmune diseases such as rheumatoid arthritis and ankylosing spondylitis [171]. In a murine glaucoma model, the administration of etanercept has been shown to inhibit TNF-α signaling, leading to a reduction in glial activation and the preservation of RGCs [172]. In the context of biologic medications for the treatment of glaucoma, Geyer and Levo extensively reviewed the current literature regarding the autoimmune aspects of glaucoma [17]. They suggested that immunomodulatory drugs approved for autoimmune diseases, such as Janus kinase inhibitors, anti-cytokines, and rituximab (an anti-CD20 monoclonal antibody), may be suitable for managing glaucoma [17]. However, an investigation designed to test intravitreal injections of rituximab for the treatment of retinal lymphomas reported that the procedure could inadvertently lead to an elevation in IOP, necessitating the use of antiglaucoma drugs postoperatively [173].

Another possible immunomodulating strategy is to target the TGF-β2 or the NF-kB pathway. A recent study conducted on human TM cells found that baicalin, an extract from *Scutellaria baicalensis Georgi*, has the potential to prevent fibrosis by reducing the deposition of the TGF-β2-induced extracellular matrix. This effect was achieved through the modulation of the NF-kB pathway [174]. Another study utilizing an experimental mouse model of glaucoma characterized by the transgenic inhibition of astroglial NF-kB, demonstrated a protective effect against neurodegeneration in RGC axons and somas. These findings suggest potential new approaches in immunomodulation for glaucoma by targeting NF-kB, a crucial mediator of neuroinflammation [175]. However, it is important to note that NF-kB targeting may be controversial due to its essential role in regulating physiological cell survival mechanisms [129,176,177]. The lack of cell-specific NF-kB targeting can lead to severe side effects, including RGC loss, as observed in transgenic mice lacking NF-kB [178].

Modulating the complement system may also lead to a decrease in the neuroinflammation associated with glaucoma. As previously mentioned, the abnormal activation of the complement system is a known event in the pathophysiology of glaucoma, and enhanced complement activity has been observed in glaucoma models [179,180]. Building upon this knowledge, a study investigated the effect of the combined inhibition of the endothelin and complement systems in a mouse model of glaucoma, and found a significant neuroprotective impact, as 80% of the mice subjected to the treatment exhibited no detectable glaucoma [181]. Additionally, a recent study on an experimental autoimmune glaucoma model demonstrated that an intravitreal treatment with an antibody against complement factor C5 suppressed complement activation, leading to reduced RGC loss and the prevention of degenerative events associated with immune dysregulation in glaucoma [182].

Another class of compounds tested as immunomodulators for glaucoma is the group of cAMP phosphodiesterase inhibitors. Ibudilast is a non-selective 3′,5′-cyclic adenosine monophosphate (cAMP) phosphodiesterase (PDE) inhibitor with a specific affinity for PDE type 4. It possesses important anti-inflammatory and vasodilator properties and is used in the treatment of stroke and asthma [183,184]. Ibudilast has been shown to suppress glial activation and the generation of inflammatory cytokines [185]. Ocular hypertension has been found to upregulate PDE type 4 in Müller cells, the major glial cell type in the retina [186]. In a rodent model of glaucoma, Ibudilast was found to mitigate neuroinflammation and improve RGC viability through the cAMP/Protein kinase A axis [187].

Interestingly, another PDE also expressed in the retina is PDE type 5 [188]. Sildenafil, a PDE type 5 inhibitor commonly used to treat erectile dysfunction, due to its vasorelaxant effects, has been investigated in a glaucoma rodent model. The study demonstrated that sildenafil promotes RGC survival by modulating the TNF-α pathway [189]. Furthermore, sildenafil was the subject of a dedicated clinical trial (NCT04052269) aimed at evaluating the effect of PDE inhibitors on blood circulation in the retina and the choroid vessels of patients with glaucoma using OCT scans [190]. However, the trial was suspended due to the COVID-19 pandemic.

Antibiotics have been also investigated as possible immunomodulatory approaches to manage glaucoma. Minocycline is a tetracycline antibiotic that has demonstrated anti-inflammatory and vasoregulatory activities in a retinal ischemia/reperfusion model [191]. The administration of minocycline in glaucomatous rodent eyes and rodent eyes after optic nerve transection has been shown to prevent RGC loss by suppressing proapoptotic cascades [192]. Another study on a murine glaucoma model demonstrated that minocycline can mitigate glial activation and improve RGC axonal transport and integrity [193]. Similarly, in an experimental model of glaucoma, minocycline antagonized microglial reactivity, preserving RGC axons and glia from degeneration [194]. However, in a recent investigation using a glaucoma-like degenerative retinal model, minocycline was found to decrease inflammation and glial activation, but did not provide complete protection for RGCs [195]. In a rodent model of chronic OHT, intravitreal injections of minocycline-induced Müller cell autophagy and increased RGC survival, confirming its role as a microglial inhibitor [196]. Mechanistically, it has been suggested that minocycline can upregulate the genes associated with the anti-apoptotic Bcl-2 family, as observed in an optic nerve transection model, human TM cells, and optic nerve head astrocytes [197,198].

Azithromycin is a macrolide antibiotic with immunomodulatory properties that has been explored, for example, in the treatment of respiratory disorders [199]. In a rodent model of ischemia/reperfusion, the post-injury administration of azithromycin exhibited a neuroprotective effect by preventing RGC loss through the suppression of Bcl-2-associated death promoter (Bad) upregulation, and the inhibition of metalloproteinase (MMP)-2/-9 activity and the ERK1/2 pathway [200]. Consistent with these findings, another recent investigation using a rodent model of glaucoma found that azithromycin preserved RGCs from apoptosis and attenuated neuroinflammation by decreasing the Bcl-2 associated X-protein (Bax)/Bcl-2 ratio, TGF-β levels, and TNF-α levels [189].

Another method to reduce neuroinflammation in glaucoma that has been proposed is the employing of stem cell-based treatments. Therapies built on stem cells are well-known for their regenerative capabilities; however, they have also been suggested as an approach to modulate inflammatory events [129]. In this regard, several publications have highlighted the protective effect of mesenchymal stem cells (MSCs) on RGCs, inducing neuroprotection in terms of preserving RNFL thickness in a rodent optic nerve crush model [201] and in a glaucoma model [202]. Nevertheless, a dedicated clinical trial involving two patients with advanced glaucoma (NCT02330978) showed that intravitreal injections of autologous bone marrow-derived MSCs did not result in changes to the electroretinographic responses or improvements in visual acuity. In one of the patients, retinal detachment occurred two weeks after treatment [203]. These findings indicate the need for modified MSCs for glaucoma treatment [204]. A recent study examined the immunomodulatory features and safety of MSCs in an ex vivo neuroretina explant model. The study assessed the capabilities of MSCs to attenuate glial activation, TNF-α signaling, and IL-1β signaling. However, it also confirmed edema and gliosis as side effects of the stem cell treatment [205].

Toll-like receptors and microbiota may also be targeted for immunomodulation in glaucoma. TLRs can interact with lipopolysaccharides as well as with DAMPs, playing a role in glial activation signaling and the amplification of neuroinflammation [129]. Interestingly, commensal microbiota has been found to be partially involved in the pre-sensitization of T cells observed in murine infiltrates during glaucomatous neurodegeneration [206]. Astafurov et al. administered lipopolysaccharides subcutaneously in two different murine glaucoma models, resulting in increased axonal degeneration and RGC loss, along with microglial activation in the optic nerve and retina [207]. The study also demonstrated that lipopolysaccharide-induced TLR-4 activation was responsible for amplifying neuroinflammation and complement activation, thereby exacerbating glaucomatous degeneration. Naloxone, an opioid shown to inhibit TLR-4, partially attenuated these effects [207,208,209]. TAK-242, also known as resatorvid, is a small-molecule cyclohexene derivative that acts as a TLR-4 inhibitor. It has been shown to attenuate glial activation in RGCs of an optic nerve crush model by reducing the p38 pathway and NF-kB activation [210]. In addition, TAK-242 has been demonstrated to block fibroblastic proliferation of the Tenon’s capsule in a rodent model, suggesting its potential as an anti-scarring drug after glaucoma surgery [211]. Mechanistically, TAK-242 decreases the TGF-β2 pathway in human TM through TLR-4 inhibition [212].

Furthermore, short-chain fatty acids (SCFAs), products of microbiota in fermentation, have been described as mediators of microglial homeostasis and are capable of binding to TLRs [213,214]. Chen et al. revealed that SCFAs can suppress inflammatory responses in retinal astrocytes by decreasing proinflammatory cytokines, such as IL-6 [214]. Their potential role in modulating the microbiota and counteracting inflammation in the neurodegeneration of glaucoma has been suggested [17].

### 4.2. Promising Antioxidants for Glaucoma Treatment

Numerous naturally occurring molecules with antioxidant properties have been investigated in preclinical and clinical studies for their potential benefits in preserving RGCs in glaucoma.

Vitamin B3, or niacin, has been studied for its antioxidant features in the treatment of glaucoma [215]. An epidemiological study conducted in Korea found that patients with NTG had a lower dietary intake of niacin compared to other nutrients, suggesting a possible negative correlation between vitamin B3 intake and NTG risk [216]. Preclinical investigations on a murine glaucoma model have shown that the administration of nicotinamide (the amide form of niacin) is effective in preventing and slowing down the progression of glaucoma by attenuating the age-related decline of nicotinamide adenine dinucleotide (NAD) [217]. A randomized controlled trial involving 57 patients with glaucoma demonstrated that nicotinamide supplementation can improve the inner retinal function [218].

Astaxanthin (AST) is an antioxidant molecule found in microalgae and other sources [219,220]. In a rat model with an elevated IOP, AST was shown to decrease apoptotic cascades [221]. In a murine model of NTG, AST demonstrated the ability to prevent RGC loss [222]. Mechanistically, AST appears to activate the nuclear factor erythroid-derived 2-related factor 2 (Nrf2), a transcription factor that upregulates several antioxidant genes, thus attenuating RGC loss in glaucoma [223].

Resveratrol is a polyphenol present in grapes, berries, and peanuts, and is known for its antioxidant properties [224]. This molecule has been shown to activate sirtuin1 (SIRT1), a nuclear NAD^+^-dependent deacetylase that upregulates the Nrf2/ARE (antioxidant response elements) pathway [225,226]. In a rodent glaucoma model, resveratrol was reported to attenuate RGC loss [227]. Moreover, resveratrol was shown to preserve RGCs from ROS-triggered apoptosis by suppressing MAPK cascades (p38, JNK, ERK) [228]. Likewise, in a mouse model of retinal ischemia/reperfusion injury induced by an elevated IOP, resveratrol promoted RGC survival by reducing oxidative stress, possibly via the downregulation of NOX2 expression [229].

The α-lipoic acid (ALA) is found in vegetables, fruits, and the liver or heart of animals [230]. In a glaucomatous mouse model, ALA decreased ROS formation and increased the activity of antioxidant enzymes like NOS and HO-1, possibly through the activation of Nrf-2 [230]. In a prospective case–control study, a formula containing ALA and other antioxidants, including vitamin C, enhanced the systemic markers of antioxidative status, such as total antioxidant status (TAS), and reduced the systemic oxidative marker malondialdehyde (MDA), a marker of lipid peroxidation, in the blood of patients with POAG [231].

Curcumin is a constituent of the spice turmeric, traditionally used in medicine, and possesses antioxidant properties [232]. In a rodent model of chronic elevated IOP, curcumin reduced ROS generation and inhibited apoptotic pathways by downregulating proapoptotic proteins, such as caspase-3, Bax, and cytochrome c [233]. In a murine model, curcumin prevented RGC loss by blocking MAPK, caspase-9, and caspase-3 activation [234].

Flavonoids are a class of molecules present in plants that possess antioxidant properties. Plant extracts from *Gingko biloba* L. contain over 70 diverse flavonoids, which have been shown to interfere with apoptotic pathways by binding with proteins such as p53, Bax, Bcl-2, and caspase-3/-9 [235]. Flavonoids in *Gingko biloba* L. may attenuate RGC injuries in glaucoma by suppressing ROS-induced apoptosis [236]. However, a clinical study comparing oral antioxidants, including extracts of *Ginkgo biloba* and α-tocopherol, for the treatment of glaucoma (NCT01544192) did not show any clear benefits associated with the use of *Ginkgo biloba* [237]. Coenzyme Q_10_, another flavonoid, has been shown to reduce glutamate excitotoxicity and ROS formation in a mice model of glaucoma, thus preserving RGCs from apoptosis by reducing Bax expression and enhancing Bad protein expression [238]. Currently, a clinical trial (NCT03611530) is underway to determine the effect of a formula containing coenzyme Q10 and vitamin E on patients with POAG [239]. Another trial (NCT04784234) is also ongoing, testing a mixture of *Ginkgo biloba*, α-lipoic acid, coenzyme Q10, curcumin, and other naturally occurring compounds in 100 patients with POAG. The expected completion date for this study is the end of 2023.

In a recent study from our laboratory, we found that mice devoid of the M_1_ muscarinic acetylcholine receptor subtype display a reduced RGC density and elevated retinal ROS levels at an advanced age despite a normal IOP [240]. Moreover, retinal mRNA levels for the pro-oxidant enzyme NOX2 were elevated, but mRNA levels for the antioxidant enzymes SOD1, HO-1, and GPX1 were reduced, suggesting that the M_1_ receptor may play an important role in regulating ROS levels in the retina and thus in neuroprotection [240]. In support of this concept, various other studies have reported the neuroprotective effects of cholinergic agents on retinal neurons, pointing to the involvement of the M_1_ receptor [241,242,243]. Huperzine A, an alkaloid extracted and isolated from the plant *Huperzia serrata*, inhibits acetylcholinesterase activity, thus increasing acetylcholine levels. In a recent study, huperzine A was reported to produce neuroprotective effects in a rat retina subjected to ischemia/reperfusion injury, via the involvement of the M_1_/AKT/MAPK signaling pathway and by reducing oxidative stress [244]. Based on these promising studies, the role of the M_1_ signaling pathway in ROS generation and in neuroprotection in the retina should be pursued further.

In addition to naturally occurring antioxidants, several existing drugs with antioxidant properties have been investigated for their potential benefits in glaucoma.

Valproic acid (VPA), an antiepileptic drug, has been shown in a murine model of NTG to attenuate excessive ROS levels and improve RGC survival through a cascade associated with ERK [245]. In a retina explant model, VPA was found to decrease the expression of pro-inflammatory cytokines and reduce microglial activation [246]. A dedicated clinical trial demonstrated that VPA had benefits in patients in the advanced stages of glaucoma, and improved their visual acuity [247].

N-acetylcysteine, commonly used in cases of paracetamol overdose and as a mucolytic agent in respiratory diseases, possesses antioxidant capabilities [248]. It attenuated retinal oxidative stress caused by an elevated IOP when combined with brimonidine in a rodent model of OHT [249]. N-acetylcysteine has been shown to enhance concentrations of glutathione, a potent antioxidant, inhibiting oxidative stress and RGC autophagy in a mouse model of NTG [250]. Another study demonstrated that this molecule can preserve RGCs from autophagy by interfering with the HIF-1α axis via BNIP3 (Bcl2 interacting protein 3) and the PI3K/Akt/mTOR cascade [251].

Edaravone, an anti-stroke drug, possesses free radical scavenging features [252]. It has been shown to inhibit the JNK/p38 proapoptotic pathways in glaucoma models, preventing RGC loss [253,254,255].

Rapamycin, a macrolide antibiotic with anti-neurodegenerative capabilities reported in Alzheimer’s and Parkinson’s diseases, has been found to increase RGC survival in rat glaucoma models. It counters the release of TNF-α from microglia, regulates NF-kB activity, and retains Akt phosphorylation to antagonize RGC apoptosis [256,257,258].

Geranylgeranylacetone, a compound used in the treatment of gastric ulcers, possesses antioxidant properties. In the retina, it promotes the activity of thioredoxin and HSP-72, preserving it against apoptosis [259]. In a mouse model of NTG, geranylgeranylacetone counteracted RGC death by upregulating HSP-70 and reducing caspase-3 and -9 activities [260].

Metformin, a widely used antidiabetic medication, has been shown in eye drop solutions to prevent fibrosis after glaucoma surgeries in a rat model by activating the AMP-activated protein kinase (AMPK)/Nrf2 signaling pathway [261].

Valdecoxib, a selective cyclooxygenase (COX)-2 inhibitor commonly used in osteoarthritis and rheumatoid arthritis, was shown in an investigation to suppress apoptosis in ischemia/reperfusion-induced glaucoma-like damaged cells of rats by blocking the ATF4-CHOP axis [262], thereby preventing CHOP-induced ROS-formation [127]. Another compound that antagonizes ER stress is 4-phenylbutyric acid (4-PBA). Traditionally employed in cystic fibrosis since the 1990s [263,264], 4-PBA has been found to mitigate ROS formation in activated microglia [265]. It can counteract ROS formation related to a high-fat diet or acute ammonia challenge by opposing ER stress [266]. In a mouse model of glaucoma, 4-PBA demonstrated an ability to reduce ER stress and prevent disease phenotypes [267]. Another study revealed that 4-PBA could reduce IOP by activating matrix MMP-9 and subsequent extracellular matrix degradation [268].

Another class of molecules which also may have the potential for neuroprotective use is represented by target-specific synthetic compounds. These molecules represent a new frontier in combating oxidative stress in glaucoma and focus on inhibiting specific molecular targets. One promising class of compounds are the NOX inhibitors, which aim to counteract the adverse effects of glial activation and supplement traditional IOP-reducing strategies [269]. GKT137831, also known as setanaxib, is a dual inhibitor of NOX1 and NOX4. It has demonstrated beneficial effects in mitigating retinal inflammation and ischemia by reducing hypoxia-related ROS formation [270]. Another notable compound in this class is GLX7013114, a specific NOX4 inhibitor. Intravitreal injections of GLX7013114 have been effective in mitigating glial activation in a rat model of α-amino-3-hydroxy-5-methyl-4-isoxazolepropionic acid (AMPA)-induced retinal excitotoxicity [271].

NOX inhibitors offer new possibilities in the field of antioxidants for glaucoma treatment, as they act independently of IOP to counteract oxidative stress, prevent RGC loss, and attenuate neuroinflammatory events.

Another emerging class of molecules are ROCK inhibitors, as demonstrated by the approval of netarsudil. Among them, Y-27632 is a noteworthy ROCK inhibitor under investigation. This potential drug has been shown to upregulate antioxidant agents such as catalase and partially reduce ROS formation [272]. Moreover, Y-27632 induces phagocytosis in glaucomatous TM cells, leading to IOP reduction [273]. Ripasudil, also known as K-115, is another ROCK inhibitor that promotes endothelium-independent relaxation in porcine retinal arterioles while suppressing ET-1 activity, suggesting its potential as an antiglaucoma drug [274].

In summary, by targeting the ROCK pathway, these molecules hold significant potential for glaucoma treatment. They optimize TM functionality, reduce fibrotic processes, and potentially lower IOP.

## 5. Conclusions and Future Perspectives

This review has shed light on the key factors involved in the pathophysiology of glaucoma, including oxidative stress, mitochondrial dysfunction, and neuroinflammation. By focusing on recent advances and new insights, we have provided an updated understanding of the underlying processes that lead to RGC loss in glaucoma. Through a comprehensive investigation of the multifaceted glaucoma pathogenesis, this research aims to facilitate the development of new curative strategies. Immunomodulatory and antioxidant drug candidates have shown promise in preclinical studies as effective options for promoting neuroprotection and RGC survival. These therapeutic approaches offer the potential to target glaucoma independently of IOP, which is currently the primary focus of glaucoma treatment. However, the development of immunomodulatory and antioxidant therapies presents challenges. The delicate balance between pro- and anti-inflammatory events and pro- and anti-apoptotic processes must be carefully deciphered to ensure that therapeutic interventions do not have unintended detrimental effects. Maintaining the redox balance to guarantee cellular homeostasis is also critical. New potential drugs should aim to minimize the possibility of harmful side effects while enhancing neuroprotection.

Additionally, challenges related to dedicated clinical trials, such as biomarker sensitivities, long-term follow-up, and drug bioavailability, need to be addressed to bridge the existing translational discrepancy between preclinical and clinical outcomes.

Considering our in-depth exploration of glaucomatous pathophysiology and experimental investigations, we extrapolate that immunomodulatory agents and antioxidants represent a significant opportunity to augment the effectiveness of pharmacological treatments and improve patient outcomes for glaucoma. These therapeutic strategies have the potential to complement established IOP-lowering drugs and offer new avenues for enhancing overall glaucoma management. Further research and clinical trials are necessary to fully realize the potential of these emerging curative strategies in the fight against glaucoma.

## Figures and Tables

**Figure 1 pharmaceuticals-16-01193-f001:**
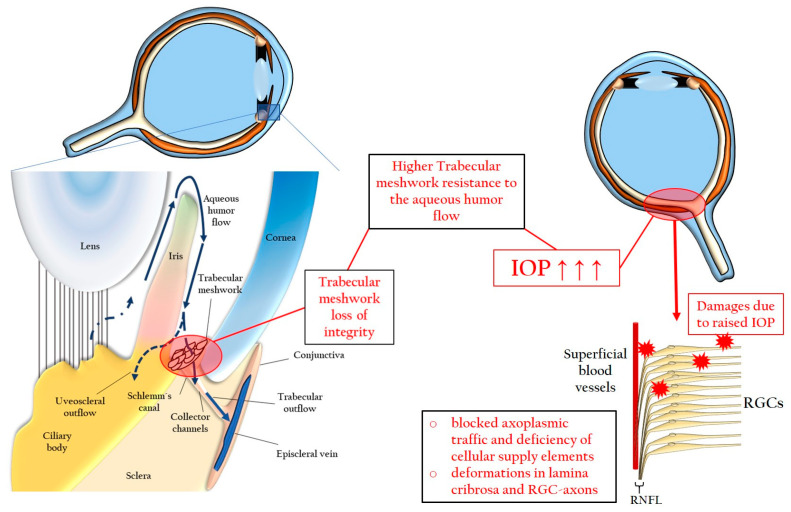
RGC-degeneration due to an elevated IOP in RGCs. IOP: intraocular pressure; RGC: retinal ganglion cell; RNFL: retinal nerve fiber layer. Up arrows mean increase.

**Figure 2 pharmaceuticals-16-01193-f002:**
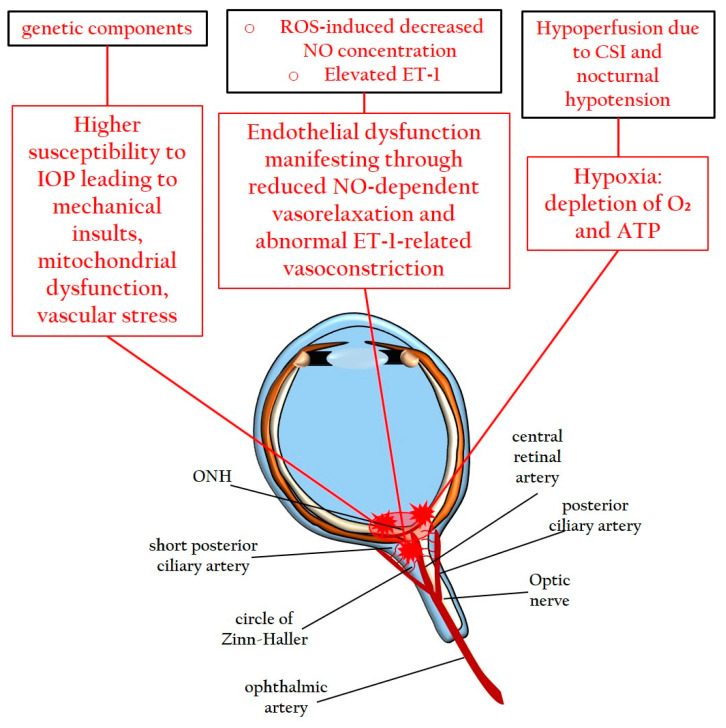
Possible etiologic initiators of NTG. IOP: intraocular pressure; ATP: Adenosintriphosphat; ONH: optic nerve head; CSI: cerebral silent infarcts; NO: nitric oxide; ET-1: endothelin-1; ROS: reactive oxygen species.

**Figure 3 pharmaceuticals-16-01193-f003:**
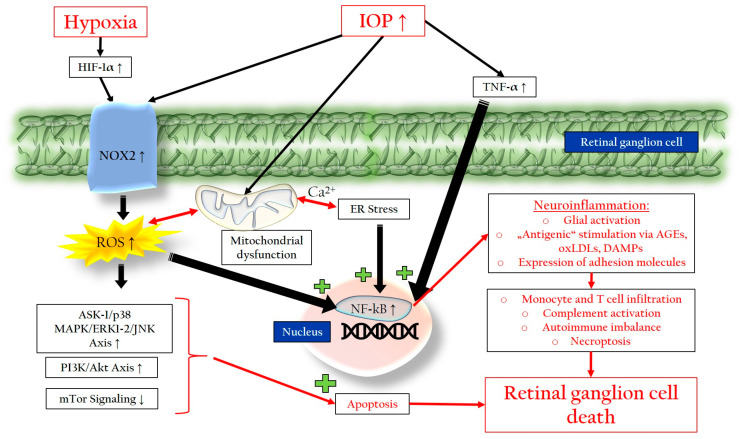
Retinal ganglion cell death in glaucoma caused by ROS-excess, NF-kB activation, complement activation, and autoimmune imbalances. IOP: intraocular pressure; NOX2: nicotinamide adenine dinucleotide phosphate oxidase type 2; NF-kB: nuclear factor ‘kappa-light-chain-enhancer’ of activated B-cells; ROS: reactive oxygen species; HIF-1α: hypoxia inducible factor 1α; TNF-α: tumor necrosis factor-α; ER: endoplasmic reticulum; ASK-1: apoptosis signal-regulating kinase 1; MAPK: mitogen-activated protein kinase; ERK: extracellular-signal-regulated kinase; JNK: Janus kinase; PI3K: Phosphoinositide 3-kinase; Akt: Ak strain transforming; mTor: mammalian target of rapamycin. Up arrows mean an increase or upregulation. Down arrows mean a decrease or downregulation.

## Data Availability

Data sharing is not applicable.

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
