# Peer review of "Immunomodulatory and Antioxidant Drugs in Glaucoma Treatment"

_pharmaceuticals, 2023, doi:10.3390/ph16091193_

Round 1

Reviewer 1 Report

Buonfiglio et al. provide a detailed description of glaucoma, its risk factors and treatment options. The manuscript would benefit from a few changes (see attached file). On top of that, I'd suggest adding a small intro section between section 3 and 3.1 as well as 4 and 4.1 to make the text more cohesive. It would also be great if you can link different paragraphs in sections 4.1 and 4.2, so they read more like a story. Now it reads like a list of items put together which are ok separately but don't read that great when put together. This can be 1-2 sentences to tie everything into something that flows nicer.

Author Response

We thank the reviewer for the comments and suggestions. According to these suggestions, we made changes in the text.

1.)        The manuscript would benefit from a few changes (see attached file)

Response to 1.) We are very grateful for these remarks. We changed the text in all parts suggested by the Reviewer (changes in the text are underlined)

2.)        I'd suggest adding a small intro section between section 3 and 3.1 as well as 4 and 4.1 to make the text more cohesive

Response to 2.) We are very grateful for this comment. We added an intro between 3 and 3.1 (lines 154-156, changes are underlined) and 4 and 4.1 (lines 395-401, changes are underlined).

3.)        It would also be great if you can link different paragraphs in sections 4.1 and 4.2

Response to 3.) We are very grateful for this suggestion. We now linked every subheader in 4.1 (changes are underlined) and 4.2 (changes are underlined).

Reviewer 2 Report

This review described the inflammatory and prooxidant mechanisms in the pathophysiology of glaucoma, and introduced the immunomodulatory and antioxidant candidates for glaucoma. There are a few suggestions to improve the manuscript.

1.     The first few chapters of the full article, such as “General Characteristics of Glaucoma”, are textbook or guide content, and it is recommended to simplify.

2.     The content since Chapter 3.2 is closely related with the topic of this review. Over the past decade, there have been significant advances in the knowledge of inflammation, oxidative stress and mitochondrial dysfunction in the course of glaucoma. However, the description is too general and lacks organization, and it is recommended to further subdivide.

3.    In glaucoma research, targeting aqueous humor production and outflow pathway, and RGC and neuroprotection, are two very different areas. It is better to make a clear distinction in the introduction of related drug candidates.

4.      The introduction of some contents should be updated with recent literature. For example, Line231-233, the literature cited is from 2003, and there have been many advances in understanding Optineurin in recent years.

Author Response

We thank the reviewer for the comments and suggestions. According to these suggestions, we made changes in the text.

1.)        The first few chapters of the full article, such as “General Characteristics of Glaucoma”, are textbook or guide content, and it is recommended to simplify.

Response to 1.) Thank you for your remark. We fully simplified chapter 2 (sections 2.1, 2.2, 2.3). Please see our changes in the text (lines 48-152, changed passages are underlined).

2.)        The content since Chapter 3.2 is closely related with the topic of this review. Over the past decade, there have been significant advances in the knowledge of inflammation, oxidative stress and mitochondrial dysfunction in the course of glaucoma. However, the description is too general and lacks organization, and it is recommended to further subdivide.

Response to 2.) According to the reviewer’s suggestion, we subdivided chapter 3.2 in more subheader (3.2.1 to 3.2.5, lines 250-384, changed text is underlined).

3.)        In glaucoma research, targeting aqueous humor production and outflow pathway, and RGC and neuroprotection, are two very different areas. It is better to make a clear distinction in the introduction of related drug candidates.

Response to 3.) Thank you for your recommendation. We added statements in the introduction of drug candidates (chapter 4, lines 395-399, changes are in text underlined).

4.)        The introduction of some contents should be updated with recent literature. For example, Line231-233, the literature cited is from 2003, and there have been many advances in understanding Optineurin in recent years.

Response to 4.) We are very grateful for this comment. We updated the literature concerning Optineurin (new references: 66-73, lines 206-214, changes in the text are underlined). Moreover, we supplemented the text, especially in subheader 3.2.1. , 3.2.2. and 3.2.4. with updated references (new references: 92; 104; 110-118; 139-141).

Reviewer 3 Report

1- Grammar needs to be checked.

2- Figure 1 has no reference.

3- Section 2.3 should be renamed to "Pharmaceutical Approaches in Treatment and Surgical Interventions."

4- The prognosis of glaucoma should be added.

5- The conclusion should not contain references and should only reflect the author's opinion. It needs to be rewritten from scratch.

6- The references need to be updated.

1- Grammar needs to be checked.

Author Response

1.)        Grammar needs to be checked.

Response to 1.) Thank you for your suggestion. We revised the full article to improve the grammar.

2.)        Figure 1 has no reference.

Response to 2.) We deleted Figure 1.

3.)      Section 2.3 should be renamed to "Pharmaceutical Approaches in Treatment and Surgical Interventions."

Response to 3.) Thank you for your remark. As you suggested, we changed the name of the section 2.3.

4.)        The prognosis of glaucoma should be added.

Response to 4.) We are very grateful for this comment. We added a short paragraph upon the prognosis of glaucoma at the end of section 2.1 (lines 81-88, changes in the text are underlined).

5.)        The conclusion should not contain references and should only reflect the author's opinion. It needs to be rewritten from scratch.

Response to 5.) We are grateful for this recommendation. We rewrote conclusion (lines 696-721, text is underlined).

6.)        The references need to be updated.

Response to 6.) We updated several references in section 3.1. (new references: 66-73) and 3.2., and especially in subheader 3.2.1., 3.2.2. and 3.2.4. (new references: 92; 104; 110-118; 139-141) with recent publications regarding the glaucomatous pathophysiology (changes in the text are underlined).

Round 2

Reviewer 2 Report

The authors have made most of recommended changes.